# An Audit of the Nutrition and Health Claims on Breakfast Cereals in Supermarkets in the Illawarra Region of Australia

**DOI:** 10.3390/nu11071604

**Published:** 2019-07-15

**Authors:** Romi L. Sussman, Anne T. McMahon, Elizabeth P. Neale

**Affiliations:** 1School of Medicine, University of Wollongong, Wollongong 2522, New South Wales, Australia; 2School of Health and Society, University of Wollongong, Wollongong 2522, New South Wales, Australia; 3Illawarra Health and Medical Research Institute, University of Wollongong, Wollongong 2522, New South Wales, Australia

**Keywords:** public health, nutrition labelling, nutrition claims, health claims, food marketing, nutrient profiling

## Abstract

Nutrition and health claims can promote healthier food choices but may lead to consumer confusion if misused. Regular monitoring of claims is therefore required. This study aimed to explore the prevalence of nutrition and health claims carried on breakfast cereals in supermarkets, and to assess claim compliance with regulations. Nutrition and health claims on breakfast cereal products across five supermarkets in the Illawarra region of New South Wales, Australia, were recorded in a cross-sectional audit. Prevalence of claim type and claim compliance was determined. Claims were compared across categories of breakfast cereal. Almost all (95.7%) products audited carried at least one nutrition or health claim; nutrition content (*n* = 1096) was more prevalent than health claims (*n* = 213). Most claims (91.6%) were compliant with regulations. Additionally, claim prevalence and type differed according to breakfast cereal category, with the highest proportion of claims appearing on ‘health and wellbeing’ and ‘muesli’ products. There is a high prevalence of nutrition and health claims on breakfast cereals, with most claims compliant with regulations. Research should investigate consumer interpretation of claims and the impact of applying nutrient profiling for all claims to assist consumers to make informed health choices.

## 1. Introduction

Obesity is a growing issue [1] and currently 63% of Australian adults are overweight or obese [2]. Globally, there is a shift towards energy-dense diets [3], with poor nutrition, and nutrition-related risk factors are accountable for rises in obesity and diet-related chronic diseases [1,4,5,6]. The food environment is an acknowledged contributor to the obesity epidemic, given its potential influence on health behaviours [5,7,8], with food retailers and supermarkets a large component of the food environment [5,9]. Supermarkets are recognised for improving the availability, quality and diversity of fresh food, yet have a powerful influence on food choice through product pricing, placement and advertising [10,11,12].

The availability of nutritional information is essential to enable and empower consumers to be in greater control of their diet [13,14]. Nutrition and health claims refer to the information on packaged food products regarding nutrient content and associated health impacts [15]. These claims may increase consumer awareness of the health characteristics of foods and promote healthier food choices. While claims are intended to benefit individuals and limit consumer misguidance, food manufacturers may use claims as marketing techniques [16,17]. This may be deceiving as consumers gauge food products with claims as healthier than those that do not have claims, otherwise known as the ‘halo effect’ [14,16,18,19,20].

In Australia and New Zealand, Food Standards Australia New Zealand (FSANZ) govern the Food Standards Code (FSC) [21]. The role of FSANZ and other relevant concepts are summarised in Appendix A. Standard 1.2.7 specifically outlines regulations regarding food labelling, claims and advertising [15]. Prior to the release of this standard in 2013 (and its enforcement in 2016), there was no mandatory code that governed all food claim regulations [22]. Standard 1.2.7 outlines that a claim refers to an ‘implied statement, representation, design or information in relation to a food or property of food’ [15]. There are several levels of claims allowed under Standard 1.2.7, as shown in Figure 1. Nutrition content claims refer to the presence or absence of a food property, whereas health claims refer to diseases or biomarkers of diseases [15]. Standard 1.2.7 outlines requirements of products making nutrition and health claims. For example, products must meet certain qualifying criteria to make claims (for instance, products claiming to be a good source of fibre must contain at least four grams of fibre per serve). In addition, products making health claims must meet the Nutrient Profiling Scoring Criterion (NPSC), a tool which categorises foods according to their nutritional composition [15]. Application of the NPSC means that only products deemed to be “healthy” may make health claims.

Understanding and interpretation of claims amongst consumers may be difficult, particularly when distinguishing between types of claims [18,19,23,24,25,26,27,28,29,30]. Furthermore, given the rapidly changing food supply, it is important that the use and compliance of claims are regularly monitored. Since Standard 1.2.7 became mandatory in 2016, there has been limited Australian research exploring nutrition and health claim compliance. Pulker et al. [31] explored nutrition and health claims made on a selection of ultra-processed foods from large manufacturers in Australia. However, further research is required to assess the claims made on food items in supermarkets. Research highlights that breakfast cereals are amongst the food categories with the highest use of nutrition and health-related claims [12,23,32,33,34,35]. Regular breakfast cereal consumption is correlated with improved overall diet quality, lower risks of being overweight or obese and non-communicable diseases [36,37]. However, a 2011 Australian and a 2013 New Zealand survey revealed that 17 and 26% of breakfast cereals carrying claims did not meet the NPSC [23,34]. Given the prevalence of claims made on breakfast cereals, this food category provides a useful case study to explore claim use and compliance. The aim of this study was to explore the prevalence of nutrition and health claims carried on breakfast cereals within supermarkets. A secondary aim was to assess claim compliance with Standard 1.2.7. It was hypothesised that: (1) a high proportion of breakfast cereal products would carry nutrition and health claims; (2) nutrition claims would be more common than health claims; (3) a high proportion of claims will be made on products that meet the NPSC; and (4) the majority of claims would be compliant with Standard 1.2.7.

## 2. Materials and Methods

This research involved a cross-sectional audit of the nutrition and health claims made on breakfast cereal products available in a sample of supermarkets of the Illawarra region of New South Wales, Australia.

### 2.1. Pilot Methodology

Prior to conducting the audit, data collection methods were trialled in a pilot stage. To estimate product sample size, a review of supermarket product availability was conducted. It was observed that Woolworths and Coles supermarkets had similar product availability, whereas Aldi had alternate products. The sample size was estimated to be approximately 300–350 products, which aligned with samples reported in previous studies (mean *n* = 260) [23,31,34,38,39,40]. Data collection via photographs and manually recording data via standardised data collection sheets was trialled. Following piloting of data collection methods, the audit was conducted as outlined below.

### 2.2. Selection of Supermarkets and Packaged Food Products

Woolworths supermarkets were identified to have the largest proportion of supermarket shares (35.7%), followed by Coles (33.2%) and Aldi (13.2%) [41,42]. Subsequently, after piloting the methodology, three Woolworths supermarkets locations of varying socio-economic status (Bulli, Fairy Meadow and Unanderra) and Coles Wollongong and Aldi Wollongong were audited to obtain product variety [42]. All breakfast cereal products that were available at the time of data collection were audited.

### 2.3. Data Collection

Data was collected over a 10-day period in April 2018. Photographs were taken of the front, back and side labels of all breakfast cereal products. Permission was sought from store managers to photograph products and conduct the audit. All products located in the ‘breakfast cereal’ and the ‘health’ aisle were audited (with the exception of Aldi, which did not have a designated ‘health’ section).

The following information was collected for each product: brand and product name, variety, nutrient composition information from the Nutrition Information Panel (NIP), (energy, protein, saturated fat, sugar, sodium, fibre per 100 g and per serve), percentage of fruit, vegetables, nuts and legumes from the ingredient list, number, type (content, general-level health, high-level health, or therapeutic claim) and wording of claims.

If a product had multiple packaging sizes, the largest pack size was used as it was suggested that it was likely to have more claims [23]. Where the same product was sold in multiple supermarkets, the product was only included once for analysis [23,34].

### 2.4. Data Analysis

Data was entered into a Microsoft Excel v 16.9 (2017, Washington, DC, USA) spreadsheet for analysis.

#### 2.4.1. Breakfast Cereal Classification

Breakfast cereals were categorised into six categories as described previously: ‘biscuits and bites’ (e.g., Weet-Bix), ‘brans’ (e.g., bran flakes), ‘bubbles, puffs and flakes’ (e.g., corn flakes), ‘kids’ cereal’ (e.g., chocolate flavoured rice bubbles), ‘muesli’ (e.g., toasted muesli) and ‘oats’ (e.g., rolled oats) [34,38,43]. An additional breakfast cereal classification, ‘health and wellbeing’, was added to account for the ‘health and wellness’ trend [44,45]. Products which were located in a specified ‘health’ aisle were classified into the ‘health and wellbeing’ category.

#### 2.4.2. Nutrient Profiling

The FSANZ NPSC was applied to all products to assess nutritional composition and determine eligibility to carry health claims [15,46,47,48]. The NPSC allocates ‘baseline points’ for less healthy nutrients (energy, saturated fat, sugar and sodium), and allocates ‘modifying points’ for healthier nutrients (fruit, vegetables, nuts, legumes (FVNL), dietary fibre (F) and protein (P)). The final score is calculated by subtracting the ‘modifying points’ from the ‘baseline points’ [46]. Breakfast cereals are classified as ‘healthy’ if the NPSC is less than 4, and the lower the score the healthier the product is considered to be [47].

Where information required for the NPSC was not present on the NIP, it was estimated using a systematic approach. For example, as it is not mandatory to list dietary fibre on the NIP, where fibre was missing, it was estimated from similar products, as conducted in previous studies [23,34,49,50]. In these cases, the average fibre value per 100 g was calculated for each breakfast cereal category; this value was used as an estimate for products with missing values. When the percentage of FVNL was not reported it was estimated from the ingredient list, as components of a food product are listed on the ingredient list in descending order by weight [51], which also allowed FVNL content to be estimated from its position in the ingredients list in relation to other ingredients with a listed percentage. To be eligible for FVNL points a product must contain a minimum of 40% non-concentrated FVNL; 40% mixture of non-concentrated FVNL and concentrated fruit or vegetables; 25% or more of concentrated fruit or vegetables [47]. Often when quantities of FVNL were not listed, these values were not relevant as the product would not have qualified for FVNL points regardless. For example, a fruit-flavoured muesli or porridge may not provide the percentage of fruit and nuts added. However, the fruit content of such a product would be considered negligible as the oat and grain content was greater than 60% of the ingredients, meaning it would not meet the minimum requirement for FVNL points.

#### 2.4.3. Claim Type and Classification

Claims identified in the audit were classified according to Standard 1.2.7 as nutrient content, general-level health, high-level health or therapeutic claims [15]. Although Standard 1.2.7 does not specifically outline whole grain claims, they are permitted as long as the claim states that exact amount of whole grain contained in the food product. For the purpose of this study, considering the frequency of whole grain claims on breakfast cereals they were classified as nutrient content claims [15,52].

In the case where a single claim included multiple nutrients or food properties, each element was considered as an individual claim. For example, a nutrition content claim that stated ‘Contains 9 vitamins and minerals’ was classified as nine different claims, as different nutrient values and qualifying criteria were employed to assess compliance [52].

Several categories of claims were excluded from this analysis (Appendix A). Nutrition and health claims were excluded in this study if they did not explicitly pertain to Standard 1.2.7 [15]. Standard 1.2.7 does not apply to claims in a permitted Health Star Rating (HSR) Symbol [15]. Puffery claims (for example ‘no artificial colours or flavouring’, ‘preservative free’ and ‘tasty’) were excluded as they are not covered in Standard 1.2.7 [15]. Claims that stated a product was ‘organic’, ‘FODMAP friendly’, ‘paleo’, or ‘vegan’ were excluded. When wording such as ‘goodness’ or ‘nutritious’ was used to describe a product, this was not considered a claim. Additionally, the use of spokespersons or cartoon figures were not examined in this study, as these do not form part of Standard 1.2.7.

#### 2.4.4. Claim Compliance

Claim compliance was evaluated against Standard 1.2.7 [15], ensuring nutrition content claims met the ‘qualifying criteria’ for the claimed nutrient and that health claims met the ‘qualifying criteria’ and NPSC [47,52]. For example, if a product carried a nutrition claim ‘Good source of fibre’, the product would have to meet the qualifying criteria which states the product must contain minimum 4 g dietary fibre per 100 g. The Grains and Legumes Nutrition Councils’ (GLNC) Voluntary Code of Practice for Wholegrain Ingredient Content Claims [53] was used as a guide to assess compliance of whole grain claims, as has been used in previous research [31].

### 2.5. Data Quality

The reliability of claim identification and categorisation was determined by a second researcher (EN) conducting the nutrient profiling and classification and compliance of claims independently on a 10% random sample of collected data.

### 2.6. Statistical Analysis

Data was analysed descriptively using Microsoft Excel v 16.9 (2017). The average and total number of products carrying claims and the number of claims per breakfast cereal category was determined. Data was also analysed by claims—the type of claims carried in each breakfast cereal category. The number and percentage of claims compliant with regulations was calculated. The most commonly used types of claims were analysed, with less common claims grouped together for reporting. Compliance of claims was analysed by evaluating the proportion claims compliant with the NPSC and the proportion of claims compliant with Standard 1.2.7.

## 3. Results

Overall, 329 breakfast cereal products were surveyed across five supermarkets in the Illawarra region and were included in analysis. A total of 317/329 (96.4%) of products had complete data for all nutrients required for assessing claim compliance. The twelve products with incomplete data had missing fibre values which were imputed based on the average fibre value for each cereal category.

### 3.1. Proportion of Breakfast Cereal Products Carrying Nutrition and Health Claims

A total of 1309 claims were identified. Table 1 summarises the distribution of products surveyed in each breakfast cereal category, the spread of products carrying claims and the mean number of claims per product per breakfast cereal category.

Of all products surveyed, 95.7% (*n* = 315) carried some form of nutrition, health or related claim. The maximum number of claims found on any product was 16, with this product in the ‘kids’ cereal’ category. The mean number of claims per product was four (standard deviation: 3.0). The highest and lowest average number of claims per product was in the ‘brans’ (mean: 5.8, standard deviation: 3.9) and ‘oats’ (mean: 2.4, standard deviation: 0.9) categories respectively. The breakfast cereal categories which carried the largest proportion of claims were ‘muesli’ and ‘health and wellbeing’ (23.2%). ‘Kids’ cereal’ products comprised the smallest proportion of products surveyed (6.1%) yet had the second highest average of claims per product (*n* = 5.5).

### 3.2. Types of Claims Made on Breakfast Cereal Products

Table 2 shows that of the 1309 claims surveyed, 83.7% of claims were nutrition content claims and 16.3% were health claims; 15.0% were general-level and 1.3% were high-level health claims. No therapeutic claims were identified.

The most common types of claims made according to their subject classification are outlined in Appendix A. The most commonly used claims were claims classified as ‘other’ (25.7%), referring to claims not specifically outlined in Standard 1.2.7. Of all the 337 claims classified as ‘other’, 228 (67.7%) claims were whole grain-related content claims. Aside from ‘other claims’, the largest proportion of claims were source of fibre (9.2%); good source of fibre (7.3%) and source of protein (5.7%).

When examined according to claim classification, the most frequent nutrient content claims related to ‘other’ (*n* = 312), fibre (*n* = 216) and protein (*n* = 75), whilst the most common general-level health claims were fibre (*n* = 36), ‘other’ (*n* = 23) and iron (*n* = 21). Of the 17 high-level health claims made, 82.4% pertained to beta glucan claims.

### 3.3. Claim Classification According to the NPSC

The mean (standard deviation) nutrient profiling score calculated using the NPSC was 1 (4.5). Nutrient profiling scores ranged from –6 and 18. To meet the NPSC and be eligible to carry general and high-level health claims, products in this audit must have a NPSC score less than 4. Of all the breakfast cereal products audited 275 (83.6%) met the NPSC (Appendix A). ‘The cereal categories with the lowest proportion of products meeting the NPSC were the ‘bubbles, puffs, flakes’ (61.9%) and ‘kids’ cereal’ (50.0%) categories.

Most claims (84.3%) were carried on products that met the NPSC. Table 3 illustrates that 98.0% of general-level and 100.0% of high-level health claims were carried on products compliant with the NPSC. Of all nutrient content claims, 81.7% were made on products which met the NPSC, despite this not being required according to Standard 1.2.7.

### 3.4. Compliance of Claims with Standard 1.2.7

Of all claims surveyed, 91.6% (*n* = 1197) were complaint with Standard 1.2.7 including 93.6% of nutrition claims (*n* = 1024) and 81.2% of health claims (*n* = 173). Products in the ‘Health and wellbeing’ (15.2%, *n* = 46) and ‘muesli’ (13.2%, *n* = 40) categories contributed the largest proportion of non-compliant claims. Forty-one of the non-compliant nutrition content claims were ‘other’ claims and of those claims, the majority (75.6%) were whole grain-related claims (Appendix A).

Table 4 shows that general-level health claims had the poorest compliance rates in comparison with other claim types. The non-compliant general-level health claims were spread so that 50.0, 26.3 and 23.7% were carried on ‘health and wellbeing’, ‘bran’ and ‘muesli’ products, respectively. It must be noted that ambiguous wording accounted for 40.0% of non-compliant general-level health claims and were consulted with a second researcher (EN) to assess compliance. Four (10.6%) general-level health claims were non-compliant as the product did not meet the FSANZ NPSC. Whilst 11.8% of high-level health claims were non compliant, it should be noted that there were very few products with high-level health claims (1.3%).

## 4. Discussion

To our knowledge, this was the first study to systematically assess the prevalence and compliance of nutrition and health claims carried on breakfast cereals available in Australian supermarkets since Standard 1.2.7 became mandatory in 2016.

### 4.1. Proportion of Breakfast Cereal Products Carrying Nutrition and Health Claims

This study identified the high proportion of nutrition and health claims carried on breakfast cereals, with 95.7% of products carrying at least one nutrition or health claim. Results are comparable with a recent New Zealand study, which found 96.0% of breakfast cereals surveyed to carry some form of nutrition or health claim [34].

### 4.2. Types of Claims Made on Breakfast Cereal Products

This study supports findings from fAustralian and international literature that nutrition claims are more prevalent than health claims, as nutrition content claims were five times more common than health claims [23,31,34,40,54,55]. In our research, health claims represented 16.3% of all claims which is slightly higher than the proportion of health claims found in a previous Australian (14.2%), New Zealand (13.3%) and European (11.0%) study [23,34,39]. In comparison to earlier research, it appears that the frequency of health claims, particularly high-level claims has increased. This is somewhat expected as most high-level health claims were not allowed in the previous version of the FSC [32,33].

The most frequently used claims identified pertained to whole grains, fibre, protein and low sodium. This aligns with the GLNC 2016 supermarket audit of 486 breakfast cereals which found whole grains, fibre and protein to be commonly associated food properties with breakfast cereals [56]. It is plausible this is because whole grains are naturally high in dietary fibre and breakfast cereals are often promoted as a source of fibre [33,34,43], thus explaining the occurrence of claims relating to these food properties [56,57,58]. This contrasts with findings from Williams et al. [32], who identified limited whole grain claims in their 2003 audit of breakfast cereals. The growth of whole grain claims could be attributed to an increase in research on the health benefits of whole grain intake over time [59,60], highlighting how popular claims may shift over time to align with consumer interest.

### 4.3. Claim Classification According to the NPSC

The NPSC was introduced to improve labelling practices by appropriately distinguishing ‘healthier’ food products [13,50,61,62,63]. This study identified 83.6% of products met the NPSC, in contrast to a 2014 New Zealand study that illustrated 74.0% of breakfast cereals met the NPSC [34]. Of the breakfast cereal products that did not meet the NPSC in the previous study, 16.9% carried health claims [34]. In comparison, our study appears to demonstrate noteworthy progress as only four (1.9%) health claims were carried on products that did not meet the NPSC. However, 18.3% of nutrition content claims were carried on products that did not meet the NPSC. While this does not contravene Standard 1.2.7, such claims contribute to the ‘halo effect’ and may result in consumers misjudging the health value of products [12,14,16,18,19]. Hughes and colleagues [23] proposed that enabling unhealthy product to carry nutrition content claims could breach the Australian Competition and Consumer Regulations 2010 [64].

Furthermore, research suggests that applying nutrient profiling to all products carrying claims may encourage food product reformulation [28]. Given the potential for nutrient content claims which do not meet the NPSC to promote unhealthy foods, application of the NPSC to all products seeking to make claims should be considered.

### 4.4. Compliance of Claims with Standard 1.2.7

Regular assessment of compliance of all nutrition and health claims is required to avoid misinforming consumers. Most claims in this study were found to be compliant with Standard 1.2.7, although general-level health claims had the largest proportion of non-complaint claims, highlighting the need for strict monitoring of health claims.

Nutrition claims had a greater degree of compliance (93.6%) in comparison to health claims (81.2%). It should be noted that unlike health claims, products carrying nutrition claims do not need to meet the NPSC. As a result, a nutrition content claim does not ensure the product is a healthy choice.The results of this study contrast with those of Pulker et al., who found greater compliance with health claims (78.9%) than nutrition claims (17.8%) [31]. Such differences in findings could be attributed to differences in methodology; for example, Pulker et al. included other forms of packaging information (health star rating, health endorsements, marketing claims) and classified claims and packaging information using a different taxonomy. Pulker et al. [31] also focused specifically on ultra-processed foods (cereals, confectionaries, snacks, beverages), which differed from the focus on breakfast cereals in the present study.

In this study, whole grain-related claims were the most common type of nutrition content claim. This warrants further discussion, given that Standard 1.2.7 does not cover whole grain claims and subsequently the GLNC Voluntary Code was established [53]. Due to variations in the two codes, products advertising whole grains can vary largely on the whole grain content and therefore contribute to unclear messages. Current research suggests that Australian consumers are aware of whole grain benefits but experience difficulties in identifying products and quantities to consume [65,66,67,68] and do not meet recommended consumption targets [69,70]. It was proposed that better labelling and education would assist consumers to identify whole grain products [66]. It is advised that whole grain claims regulations be incorporated into Standard 1.2.7 to ensure claim consistency with less room for interpretation, promote whole grain intake and enable informed food choices.

### 4.5. Claim Prevalence, Type and Compliance According to Breakfast Cereal Categories

Understanding the types of products which carry health claims may aid the targeting of monitoring strategies in the future. Consistent with other findings, a large proportion of products in the ‘bubbles, puffs and flakes’ and ‘kids’ cereal’ categories had poor nutrition profiles according to the NPSC [31,34,38]. While half of ‘kids’ cereal’ products were considered ‘less healthy’, ‘kids’ cereal’ had the second highest proportion of health claims per cereal category. These cereal categories carried numerous nutrition content claims and had high compliance rates, exemplifying how less healthy products can carry nutrition claims which may be misleading for consumers.

Marketing techniques and promotion of unhealthy food products has been shown to increase children’s dietary intake and poor dietary choices [16,31,71,72]. Food advertising and marketing incentives are often targeted at children’s products suggesting action is required to prevent misleading marketing techniques [31,34,40,54]. These results highlight the extent of the ‘halo effect’, how claims may be used to market products in susceptible populations such as children, which is worrying as it promotes unhealthy food choices. Thus, these products and their associated claims require close monitoring.

Interestingly, products in the ‘Health and wellbeing’ category carried the largest proportion of non-compliant claims. Consumers often associate claim credibility with perceived healthiness of the food product [26,73,74], thus non-compliant claims on ‘health and wellbeing’ products heightens consumer vulnerability as they are subject to misinformation [18,24,75,76]. This may add to consumer confusion regarding health claims and in turn impedes the promotion of a healthy food environment.

### 4.6. Limitations

When considering the results of this study, it should be noted that findings from this study are not representative of all claims presented to Australian consumers as only one food category was surveyed. However, breakfast cereals were audited as they were likely to have a high occurrence of claims [19,23,32,33]. Thus, it is reasonable to assume that issues regarding claims in this study may occur in other food categories. All data values collected were reliant on the accuracy of the information presented on the NIP to assess claim compliance. Furthermore, as some products did not specify fibre and fruit, vegetable, nut, and legume content, these needed to be estimated for application of the NPSC. Although this had the potential to introduce errors in the NPSC calculation, this risk was mitigated by estimated values based on similar products and other information available on the product. Whilst this study acknowledges product availability and advertising have an influential effect on consumer food choices, such data was not specifically measured.

This research specifically explored nutrition and health claim prevalence and compliance in regard to the Standard 1.2.7. Thus, not all forms of food labelling, marketing and advertising were accounted for such as ‘Health Star Ratings’, ‘puffery’ claims, and the use of cartoons and images, which are common food marketing technique. We acknowledge this data may contribute to the ‘halo effect’ and overall perception of product ‘healthiness’.

Standard 1.2.7 does not prescribe wording for claims [15]. In the present study, the assessment of claim compliance was based on subjective decisions surrounding wording used, although terminology was discussed with the research team to determine consensus. Unspecific wording of claims has previously been noted to be an issue in determining claim compliance [31]. As a result, claim classification and compliance may be interpreted differently by researchers, the food industry and manufacturers. Thus, claim assessment, classification and compliance in this study may differ in comparison with other research. These findings do however support literature which highlights that wording and context are key factors in ensuring claims do not mislead consumers [77,78,79,80]. Claims must be readily comprehendible for consumers, therefore mandatory guidelines regarding claim wording and contexts are recommended to reduce discrepancies and the misinterpretation of claims.

## 5. Conclusions

This study demonstrates that the breakfast cereals audited had a high prevalence of nutrition and health claims, with nutrition claims being the most dominant. Most claims were compliant with regulations. However, general-level health claims had the poorest compliance and half of these claims were carried on ‘health and wellbeing’ products. It is concerning that 18.3% of nutrition content claims appeared on products that did not meet the NPSC, as consumers find it challenging to differentiate between types of claims [14,18,75] and consumers may ascribe additional benefits to these products. This may be particularly relevant to products marketed at children, which carried a high number of nutrition claims despite many products not meeting nutritional profiling cut-offs. Further research is now required to explore how different claims may be interpreted by consumers, as well as the impact of applying nutrient profiling for all products carrying any form of claim to reduce potentially deceptive labelling. Given the high occurrence of whole grain claims in this food category, mandatory whole grain regulations should be considered as part of Standard 1.2.7. Furthermore, further guidance around claim wording should be considered to avoid ambiguous and inconsistent health messages, which in turn limits the value of nutrition and health claims in supporting a healthy food environment.

## Figures and Tables

**Figure 1 nutrients-11-01604-f001:**
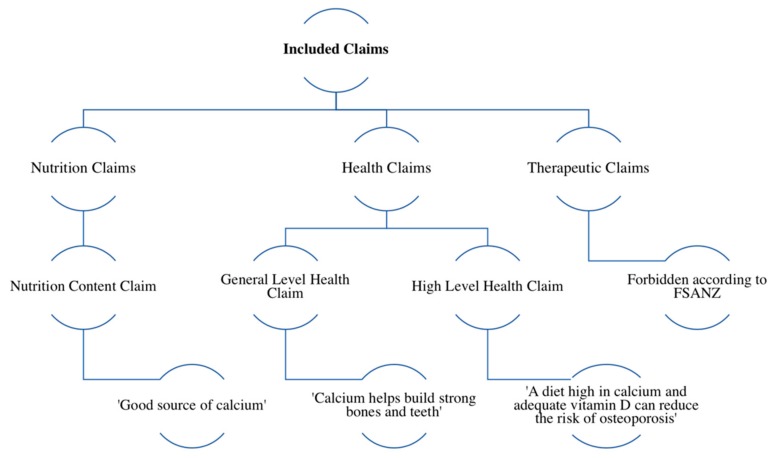
Examples and summary of claims included in this study as defined by the Food Standards Code (FSC).

**Table 1 nutrients-11-01604-t001:** Prevalence of nutrition, health and related claims across breakfast cereal categories within the supermarket audit.

Breakfast Cereal Category	Products Surveyed		Products Carrying Claims		Total Count of Claims		Average Claims Per Category
	(*n*)	% of Total Products Surveyed	(*n*)	% of Products Per Category Carrying Claims	(*n*)	% of All Claims Found	(*n*)
Biscuits and Bites	32	9.7	32	100.0	153	11.7	4.8
Brans	40	12.2	39	97.5	232	17.7	5.8
Bubbles, Puffs, and Flakes	21	6.4	17	81.0	93	7.1	4.4
Health and Wellbeing	60	18.2	60	100.0	303	23.2	5.1
Kids Cereal	20	6.1	15	75.0	110	8.4	5.5
Muesli	108	32.8	104	96.3	303	23.2	2.8
Oats	48	14.6	48	100.0	115	8.8	2.4
**Total**	**329**	**100.0%**	**315**	**-**	**1309**	**100.0%**	**4.0**

**Table 2 nutrients-11-01604-t002:** Number and proportion of claim types found to be carried across breakfast cereal categories in the supermarket audit.

Breakfast Cereal Category	Claim Type					
	Nutrition Content		General-Level Health		High-Level Health	
	Number of Claims	% of Claims Per Cereal Category	Number of Claims	% of Claims Per Cereal Category	Number of Claims	% of Claims Per Cereal Category
Biscuits and Bites (*n* = 153)	138	90.2	15	9.8	0	0.0
Brans (*n* = 232)	115	49.6	114	49.1	3	1.3
Bubbles, Puffs and Flakes (*n* = 93)	93	100.0	0	0.00	0	0.0
Health and Wellbeing (*n* = 303)	264	87.1	33	10.9	6	2.0
Kids Cereal (*n* = 110)	91	82.7	19	17.3	0	0.0
Muesli (*n* = 303)	284	94.1	15	5.0	4	1.3
Oats (*n* = 115)	111	96.5	0	0.0	4	3.5
**Total (*n* = 1309)**	**1096**	**-**	**196**	**-**	**17**	**-**

**Table 3 nutrients-11-01604-t003:** Nutrition and health claims carried on products in the supermarket audit according to product classification with the NPSC.

Claim Classification	Number of Claims on Products That Met NPSC	% of Claims on Products That Meet NPSC	Number of Claims on Products That Do Not Meet NPSC	% of Claims on Products That Do Not Meet NPSC
Nutrition content claims (*n* = 1096)	895	81.7%	201	18.3%
General-level health claims (*n* = 196)	192	98.0%	4	2.0%
High-level health claims (*n* = 17)	17	100.0%	0	0.0%
**Total (*n* = 1309)**	**1104**	**84.3%**	**205**	**15.7%**

**Table 4 nutrients-11-01604-t004:** Number and proportion of claims compliant with regulations; Food Standards Australia New Zealand (FSANZ) Standard 1.2.7 and the The Grains and Legumes Nutrition Council (GLNC) Code.

Claim Type	Number of Compliant Claims (*n*)	% of Claim Type That Were Compliant	Number of Non-Compliant Claims (*n*)	% of Claim Type that Were Non-Compliant
Nutrition content claim (*n* = 1096)	1024	93.6%	70	6.4%
General-level health claim (*n* = 196)	158	80.6%	38	19.4%
High-level health claim (*n* = 17)	15	88.2%	2	11.8%
**Total (*n* = 1309)**	**1197**	**91.6%**	**110**	**8.4%**

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
