# Peer review of "An Audit of the Nutrition and Health Claims on Breakfast Cereals in Supermarkets in the Illawarra Region of Australia"

_nutrients, 2019, doi:10.3390/nu11071604_

Round 1
Reviewer 1 Report
The authors took into account most of the comments and improved the manuscript as suggested.
Minor comments:
- Table 1 : I think it will help readers not familiar with Australian regulation. Despite this, I am not sure it is pertinent with a review rather than with an original article. Thus, I believe it should be moved in the Supplementary material
- English is fine but it can be revised throughout the manuscript. For instance:
o Lines 126-131: in these few lines the wording “Data was collected” has been used four times
o Line 162: check ”The lower the score, the healthier the product is considered”
- Line 156: NPSC has been already spell out
Author Response
The authors took into account most of the comments and improved the manuscript as suggested.
Thank you for this feedback
Minor comments:
- Table 1 : I think it will help readers not familiar with Australian regulation. Despite this, I am not sure it is pertinent with a review rather than with an original article. Thus, I believe it should be moved in the Supplementary material
In line with this comment, we have moved Table 1 to the Supplementary Material, and updated the table numbers in text.
- English is fine but it can be revised throughout the manuscript. For instance:
o Lines 126-131: in these few lines the wording “Data was collected” has been used four times
We have revised this section of the text to read: "Data was collected over a 10-day period in April 2018. Photographs were taken of the front, back and side labels of all breakfast cereal products. Permission was sought from store managers to photograph products and conduct the audit. All products located in the ‘breakfast cereal’ and the ‘health’ aisle were audited (with the exception of Aldi, which did not have a designated ‘health’ section).
The following information was collected for each product: brand and product name, variety, nutrient composition information from the NIP, (energy, protein, saturated fat, sugar, sodium, fibre per 100g and per serve), percentage of fruit, vegetables, nuts and legumes from the ingredient list, number, type (content, general level health, high level health, or therapeutic claim) and wording of claims." (lines 118 - 127 in revised manuscript)
o Line 162: check ”The lower the score, the healthier the product is considered”
This line has been reworded to: "Breakfast cereals are classified as ‘healthy’ if the NPSC is less than 4, and the lower the score the healthier the product is considered to be" (lines 148 - 150 in revised manuscript.- Line 156: NPSC has been already spell out
This has been amended so that NPSC is spelled as an abbreviation only (line 144)
Reviewer 2 Report
I have now read your revised manuscript in light of my earlier comments. I am satisfied both with the revisions made and also with the arguments put forward by the authors in the instances where revisions have not been made in accordance with recommendations. I am now happy to see this manuscript published.
Author Response
Thank you for this feedback